# Research on the cooperative mechanism of government and enterprise for basin ecological compensation based on differential game

Hao Sun[1]*, Guangkuo Gao[1], Zonghuo Li[2]

1 Business School, University of Shanghai for Science and Technology, Shanghai, China, 2 School of Politics and Public Administration, Soochow University, Suzhou, China

* 705143946@qq.com

## Abstract

Ecological compensation is an important means of basin pollution control, the existing researches mainly focus on the government level ignoring the important role of enterprises. Therefore, this paper introduces enterprises into the process of ecological compensation. Firstly, suppose the ecological compensation system composed of government and enterprises, the government is in the dominant position. The ecological compensation input of the government and enterprise will produce social reputation, and the ecological compensation of enterprise will also produce advertising effect. Consumer demand will be affected by social reputation and advertising effect. Then, the compensation strategies of the government and enterprise are analyzed by constructing the differential game model. The research shows that under certain conditions, the cost-sharing mechanism can realize the Pareto improvement of the benefits of government, enterprise and the whole system. Under the cooperative mechanism, the benefit of the government, enterprise and the whole system is optimal. Finally, the validity of the conclusion is verified by case analysis, and the sensitivity analysis of the relevant parameters is carried out. The conclusion can provide reference for government to establish sustainable watershed ecological compensation mechanism.

## 1 Introduction

Basin plays an important role in economic and social development as important connectors between nature and human activities [1]. With the development of science and technology, human's ability to develop and utilize natural resources has been enhanced, causing serious damage to the ecological environment of the river basin [2]. Due to the long-term accumulation of pollutants, the pollution of water resources in the basin has exceeded its self-purification capacity. It leads to frequent natural disasters, which seriously affect the survival and development of human beings [3]. Therefore, it is an urgent problem for all countries in the world to make ecological compensation for the watershed and gradually restore the ecological function of the watershed [4, 5].

**Data Availability Statement:** Because the ecological compensation mechanism proposed in this paper is a new operations management theory, direct real data is not available at present. Therefore, the parameters in this paper are

obtained from the relevant data in China Environmental Statistics Yearbook for accounting. The data in the China Environmental Statistics Yearbook are open access. Specific data sources are as follows: China Environmental Statistics Yearbook (2020) https://navi.cnki.net/KNavi/YearbookDetail?pcode=CYFD&pykm=YHJSD&bh= Second, water environment; Tenth, environmental investment China Environmental Statistics Yearbook (2019) https://navi.cnki.net/KNavi/YearbookDetail?pcode=CYFD&pykm=YZGHW&bh=N2020070223 Environmental Planning Institute of the Ministry of Ecology and Environment: Ecological compensation and biodiversity conservation Suzhou Yearbook (2020) https://navi.cnki.net/KNavi/YearbookDetail?pcode=CYFD&pykm=YSZNJ&bh= Selections of documents: The Office of the Suzhou Municipal Committee of the Communist Party of China (CPC) and the Office of the Suzhou Municipal People's Government issued the Notice on the Implementation of Opinions on Comprehensively Promoting and Implementing the Experience of the Pilot Ecological Compensation Mechanism in the Xin 'an River Basin Finance Yearbook of Anhui Province (2020) https://navi.cnki.net/KNavi/YearbookDetail?pcode=CYFD&pykm=YAHCZ&bh= Support the battle against pollution: To support horizontal ecological compensation in the upper and lower reaches of the Xin 'an River Basin; Promote ecological compensation for water environment in Dabie Mountain Area; establish and improve mechanisms for compensating for ecological damage. fiscal management: Anhui innovates to implement different types of ecological compensation mechanisms Yearbook of Lu Quan (2020) https://navi.cnki.net/KNavi/YearbookDetail?pcode=CYFD&pykm=YLUQU&bh= Ecological poverty alleviation: ecological compensation to increase income Financial Yearbook of Zhejiang Province (2019) https://navi.cnki.net/KNavi/YearbookDetail?pcode=CYFD&pykm=YZJCZ&bh=N2020010090 Research Report Selection: Quzhou City's Practice and Reflection on Establishing the Whole Urban Upstream and Downstream Ecological Compensation Mechanism Hefei Yearbook (2019) https://navi.cnki.net/KNavi/YearbookDetail?pcode=CYFD&pykm=YHFNJ&bh=N2021040077 Ecological construction and environmental protection: Chaohu Lake management Guiyang Yearbook (2019) https://navi.cnki.net/KNavi/YearbookDetail?pcode=CYFD&pykm=YPAKF&bh=N2019120318 Water environmental management: ecological compensation for water pollution control Xuancheng Yearbook (2019) https://navi.cnki.net/KNavi/YearbookDetail?pcode=CYFD&pykm=

Because the basin flows through a wide range and involves a large number of stakeholders, the pollution caused by any stakeholder to the basin is easily transferred to other regions through water flow [6]. Therefore, the pollution of a river basin usually affects multiple regions and becomes a trans-regional pollution problem [7, 8]. In order to solve this problem, countries around the world have tried various measures, such as formulating a legal system for watershed ecological protection [9], developing green energy industry [10]. Although these measures can reduce the pollution of the basin to a certain extent, it is difficult to fundamentally solve the problem of ecological environmental pollution, and then form a good feedback mechanism. Therefore, the concept of ecological compensation was put forward, namely the ecosystem service payment mechanism. It is an institutional arrangement that aims at protecting and utilizing ecological resources, adjusts the interests of stakeholders by economic means, and promotes the active protection of the ecological environment by all parties [11]. It provides a new way to coordinate the relationship between different stakeholders in the basin and solve the trans-regional water pollution problem [12, 13].

Watershed ecological compensation needs a large amount of funds. If only the payment is made by the government finance, it will cause great pressure on the government finance, and it is difficult to realize the sustainable compensation. Enterprise are the main cause of river basin pollution and the main beneficiary of river basin pollution. According to the principle that whoever damages shall restore and who benefits shall compensate, the enterprise shall bear the responsibility of compensation. Therefore, it is of great significance to bring polluting enterprises into the research framework of ecological compensation, and to discuss the decision-making behavior and influencing factors of government and enterprises in the process of ecological compensation.

As profit-oriented subjects, enterprise will weigh the cost and benefit of ecological compensation when they choose the investment level of ecological compensation. In order to encourage enterprises to actively participate in ecological compensation, government will choose to share part of the compensation costs of enterprises. The strategic choice of government and enterprise has the characteristics of long-term and dynamic. Differential game theory originated from the research on the pursuit of two parties in military confrontation carried out by the US Air Force in the 1950s, it is a combination of optimal control and game theory. It studies the continuous game of multiple players in a time-continuous system, in which the players try to optimize their independent goals, and eventually reach a Nash equilibrium over time. Therefore, this paper establishes the differential game model between the government and the enterprise, and studies the ecological compensation behavior of the government and the enterprise from the dynamic perspective.

The main contributions of this paper are as follows: (1) Taking enterprise into the study of ecological compensation mechanism the deficiency of government as a single compensation subject can be made up. (2) Behavioral decisions of government and enterprise for ecological compensation are considered from the dynamic perspective of differential game model. (3) The enterprise motivation of ecological compensation is deeply analyzed from the perspective of corporate social reputation and advertising effect.

The research structure of this paper is as follows: The second part reviews the relevant literature. The third part describes the problem and the model hypothesis. In the fourth part, the differential game model is analyzed. The fifth part compares three ecological compensation mechanisms. The sixth part carries on the numerical simulation to the ecological compensation. The seventh part is the conclusion, discussing the results and making suggestions.

YAHXC&bh=N2020030245 Pollution prevention and control: water environment ecological compensation.

**Funding:** The author(s) received no specific funding for this work.

**Competing interests:** The authors have declared that no competing interests exist.

## 2 Literature review

Through literature review, we found that the current research on watershed ecological compensation is mainly divided into three parts. Research on ecological compensation standard ecological, compensation mechanism, and ecological compensation effect evaluation, among which, the study on ecological compensation mechanism is the core.

In terms of ecological compensation standard, some scholars have studied how to determine ecological compensation standard. Taking Yanqing District of Beijing as an example, Li et al. compared the accounting methods and evaluation methods of compensation standards for regional forest ecosystems, and estimated the implementation cycle of compensation standards [14]. From the perspective of river ecosystem, forest ecosystem and wetland ecosystem, Yan et al. Determined the ecological compensation standard according to the input-output correspondence [15]. Niu et al. analyzed the standard of agricultural ecological compensation based on the standard model of consistency compensation between ecosystem and ecological value [16]. The above literature mainly focus on the specific accounting methods of ecological compensation standards by government, but do not study the determination of the subject and object of ecological compensation. In the study of the effect evaluation of ecological compensation. Peng selected the ecological, economic and social development data of Huangshan City from 2011 to 2018 to quantitatively evaluate the comprehensive benefits of ecological compensation in water source area by using entropy weight method [17]. Li et al. evaluated the comprehensive benefits of ecological compensation in Xiaoqing River Basin through empirical analysis, and the research results showed that the upstream and downstream compensation mechanism was very effective in improving the ecological environment [18]. Lu et al. evaluated the benefits of the watershed service charging system, and the results showed that the system could establish the upstream and downstream coordinated watershed management policies, thus improving water quality and quantity, and making government officials more responsible for water resources management, thus reducing water pollution to a certain extent [19]. The above literature mainly focus on the effect of ecological compensation, but do not study the realization process of ecological compensation. As the core research content of ecological compensation mechanism, many scholars have done in-depth research on it. Liu et al. analyzed the achievements and existing problems of ecological compensation in the upstream and downstream of the basin, they put forward suggestions to build a collaborative, differentiated and informationized ecological compensation mechanism in the upstream and downstream of the basin [20]. Yu proposed to build a society-led compensation mechanism for water resource ecological protection and promote the transformation of water resource protection from government protection to multi-social co-governance, which is an effective way to improve the performance of water conservation [21]. Zhou et al. suggested that a diversified market and compensation mechanism for watershed ecosystem services should be established to achieve the balance between supply and demand of watershed ecosystem services guided by the maximization of ecological, economic-social goals [22]. In the above studies on ecological compensation, most of them take the government as the compensation subject, adopt qualitative research and theoretical empirical research, and fail to clearly describe the dynamic change of the compensation subject's strategy in the process of ecological compensation.

The determination of the subject and object of watershed ecological compensation is an important content in the study of ecological compensation mechanism, which is directly related to the effectiveness of ecological compensation mechanism [23]. In the research on the subject and object of watershed ecological compensation, Gao et al. take the above and downstream governments as the subject and object of ecological compensation, and study the changes of their decision-making behaviors and influencing factors in the process of watershed

ecological compensation [24]. Some scholars have brought enterprise into the category of compensation subjects, For example, under the background of green development of the Yangtze River Economic Belt, Yang et al. used evolutionary game model to study the government-enterprise cooperative compensation mechanism in the Three Gorges Basin of the Yangtze River [25]. Cw A et al. used the difference game model to study the cooperative compensation mechanism between the government and enterprise [26], but did not consider the impact of the government's sharing of the ecological compensation cost of enterprise. Therefore, in this study, we first assume that the ecological compensation behavior of the government and enterprise will attract public attention and generate a good social reputation. The ecological compensation behavior of enterprise can also produce advertising effect, and consumer demand is affected by social reputation and advertising effect. Then, we use differential game model to analyze the optimal decision of government and enterprise under three modes: no cost sharing mechanism, cost sharing mechanism and cooperative cooperation mechanism. Table 1 shows the main differences between this study and the most relevant literature.

## 3 Problem description and model hypothesis

### 3.1 Problem description

This paper takes the ecological compensation system constituted by government and enterprise as the research object to study the investment level of ecological compensation between the government and enterprise. Government departments' input in the publicity and preferential policies of ecological compensation can enhance the public's awareness of environmental protection. The ecological compensation behavior of the enterprise will arouse the public's attention to the enterprise, and then increase the social reputation of the enterprise, at the same time, it will also produce advertising effect. The purchase demand of consumers is influenced by the social reputation and advertising effect of enterprise. The government income includes the social benefits brought by the enterprise ecological compensation and the tax increase brought by the increase of enterprise sales. In order to encourage enterprises to actively conduct ecological compensation, the government will share part of ecological compensation costs (subsidies and preferential policies, etc.) for enterprise.

The behavior choice between the government and the enterprise in the compensation process constitutes Stackelberg game. The government first makes the compensation decision and the proportion of sharing, and then the enterprise makes its own ecological compensation decision according to the government in the game process. The specific decision-making process is shown in Fig 1 and the research method diagram is shown in Fig 2.

In Fig 2, $V_1^{N*}$, $V_2^{N*}$, $V_1^{D*}$, $V_2^{D*}$ represents the optimal compensation income of government and enterprise under different mode. $S_1^{N*}$, $S_2^{N*}$, $S_1^{D*}$, $S_2^{D*}$, $S_1^{C*}$, $S_2^{C*}$ represents the optimal compensation level of government and enterprise under different mode. $L^*$ represents the cost sharing ratio of the government. $V_3^{C*}$ represents the total revenue of government and enterprise.

**Table 1. Summary of the major literature review (G: Government; E: Enterprise).**

| Articles | Compensation subject | Cost sharing | Compensation motivation | Dynamic perspective |
|---|---|---|---|---|
| Liu et al. [20] | G | × | × | × |
| Zhou, Feng [22] | G | × | √ | × |
| Gao et al. [24] | G | × | × | √ |
| Yang et al. [25] | G and E | × | √ | √ |
| Cw A, Cl B [26] | G and E | × | √ | √ |
| Our paper | G and E | √ | √ | √ |

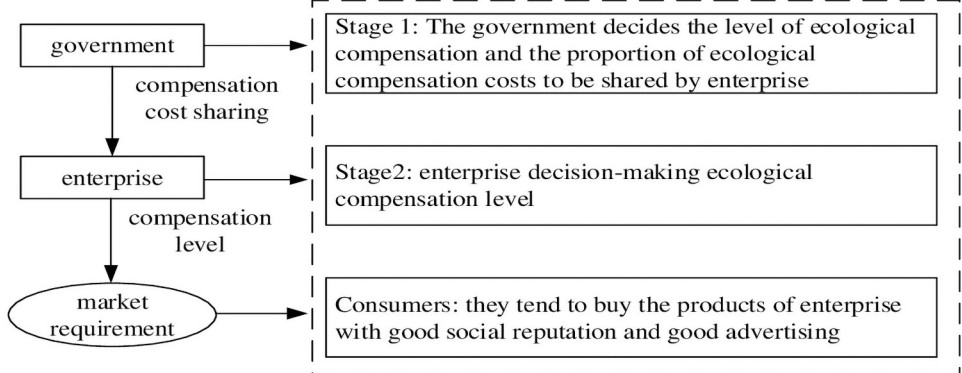

**Fig 1. Decision-making of ecological compensation by government and enterprise.**

## 3.2 Model assumptions

**Hypothesis 1** The ecological compensation cost of government and enterprises is a convex function. Referencing the hypothesis in Literature, we can be concluded that the input cost of ecological compensation by the government and enterprises at the moment $t$ is as follows:

$$C_1(t) = \frac{k_1}{2} S_1^2(t), C_2(t) = \frac{k_2}{2} S_2^2(t)$$

Where $S_1(t) > 0$ and $S_2(t) > 0$ respectively represent the ecological compensation level of the government and enterprises at the moment $t$; $C_1 > 0$ and $C_2 > 0$ respectively represent the ecological compensation input cost of the government and enterprises at the moment $t$; $k_1 > 0$ and $k_2 > 0$ represent the cost coefficients of the government and enterprises.

**Hypothesis 2** In order to encourage enterprises to invest in ecological compensation, government departments share part of the cost of enterprise ecological compensation, and the share ratio is $L(t)$, where $0 \leq L(t) \leq 1$.

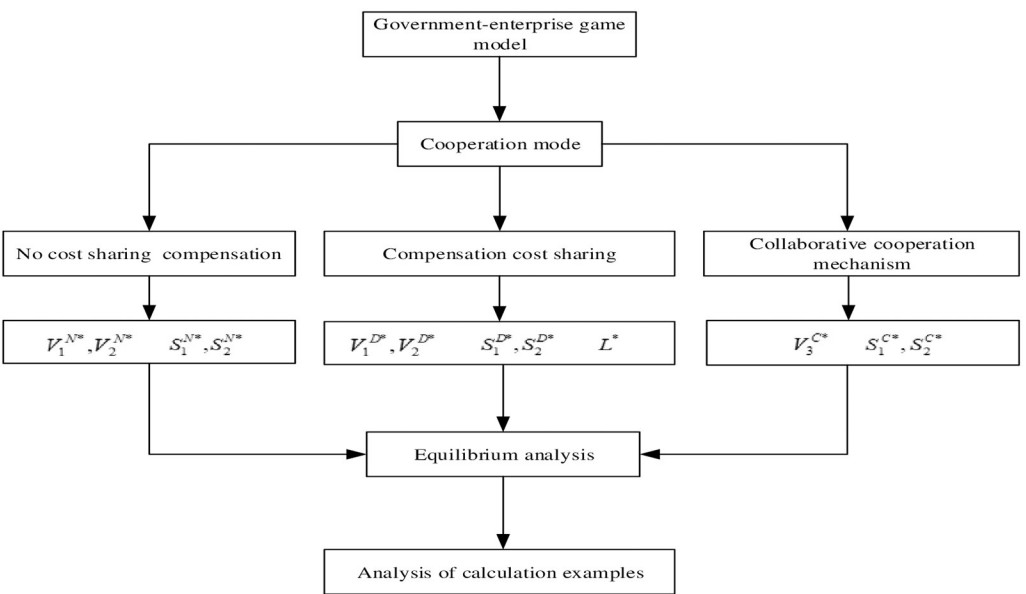

**Fig 2. Research method diagram.**

**Hypothesis 3** The ecological compensation efforts of enterprise will improve their reputation. The ecological compensation efforts government will enhance the public's awareness of environmental protection, thus increasing the public's attention to the enterprise participating in ecological compensation, and indirectly improving the reputation of the enterprise. The social reputation of enterprise is a dynamic changing process and is closely related to the ecological compensation level of the government and enterprise. Therefore, the change of the social reputation of enterprise can be described by the differential equation shown in (1):

$$\dot{R}(t) = \lambda_1 S_1(t) + \lambda_2 S_2(t) - \delta R(t) \tag{1}$$

Where $R(t)$ represents the social reputation of the enterprise at time $t$, $\lambda_1 > 0$ and $\lambda_2 > 0$ respectively represent the influence coefficient of the ecological compensation level of the government and the enterprise on the social reputation of the enterprise, $\delta > 0$ represents the natural attenuation coefficient of the social reputation of the enterprise, and $\dot{R}(t)$ represents the change rate of the social reputation of the enterprise over time $t$.

**Hypothesis 4** Consumers have green preference psychology, and their purchasing behavior is influenced by the advertising effect and social reputation of enterprises. Assume that the demand function is as follows:

$$Q(t) = \alpha S_2(t) + \theta R(t) \tag{2}$$

Where $\alpha$ represents the coefficient of the advertising of ecological compensation effort level on consumer demand, $\theta$ represents the coefficient of the influence of the social reputation of the enterprise on consumer demand.

**Hypothesis 5** The goal of both government and enterprise is to achieve the maximization of revenue, and choose their own behavioral strategies based on revenue maximization. Using the optimal control principle of differential game, the government and enterprise revenue functions can be set as follows:

$$\max_{S_1(t),L} J_1 = \int_0^\infty e^{-\rho t}[\mu_1 S_1(t) + \mu_2 S_2(t) + \pi_1 Q(t) - C_1(t) - L(t)C_2(t)]dt \tag{3}$$

$$\max_{S_2(t)} J_2 = \int_0^\infty e^{-\rho t}[\pi_2 Q(t) - (1 - L(t))C_2(t)]dt \tag{4}$$

Where $\mu_1 > 0$, $\mu_2 > 0$ represent the coefficient of influence of ecological compensation level of government and enterprises on government revenue; $\pi_1 > 0$ and $\pi_2 > 0$ respectively represents the coefficient of influence of consumer demand on government tax revenue and enterprise revenue. $\rho$ is the discount rate for government and enterprise. This paper assumes that the required parameters are constants, and the time variables are omitted in the following paragraphs for writing convenience.

## 4 Model analysis

### 4.1 No cost-sharing ecological compensation mechanism

In this case, the government is the dominant player in the game, and the enterprise is the follower. Firstly, the government determines the level of ecological compensation according to the principle of maximizing benefits. Then, the enterprise decides its ecological compensation level based on the government's ecological compensation decision. Since the government does not share the ecological compensation costs of enterprises, that is $L(t) = 0$. At this time, the

revenue function of the government and enterprise can be obtained as follows:

$$\max_{S_1(t)} J_1^N = \int_0^\infty e^{-\rho t}[\mu_1 S_1(t) + \mu_2 S_2(t) + \pi_1 Q(t) - C_1(t)]dt \tag{5}$$

$$\max_{S_2(t)} J_2^N = \int_0^\infty e^{-\rho t}[\pi_2 Q(t) - C_2(t)]dt \tag{6}$$

**Proposition1** In the absence of cost-sharing mechanism, the optimal ecological compensation level of both government and enterprise is:

$$(S_1^{N*}, S_2^{N*}) = \left(\frac{\mu_1(\rho+\delta) + \lambda_1\pi_1\theta}{k_1(\rho+\sigma)}, \frac{\alpha\pi_2(\rho+\delta) + \lambda_2\pi_2\theta}{k_2(\rho+\delta)}\right) \tag{7}$$

**Proof** The optimal control method is used to solve the equation. The numerical value $V_1^N$, $V_2^N$ satisfies the HJB equation, as follows:

$$\rho V_1^N = \max_{S_1}\left\{\mu_1 S_1 + \mu_2 S_2 + \pi_1 Q - \frac{1}{2}k_1 S_1^2 + V_1^{N'}(\lambda_1 S_1 + \lambda_2 S_2 - \delta R)\right\} \tag{8}$$

$$\rho V_2^N = \max_{S_2}\left\{\pi_2 Q - \frac{1}{2}k_2 S_2^2 + V_2^{N'}(\lambda_1 S_1 + \lambda_2 S_2 - \delta R)\right\} \tag{9}$$

To solve the right end of HJB equation, the first-order condition to maximize it is:

$$(S_1, S_2) = \left(\frac{\mu_1 + \lambda_1 V_1^{N'}}{k_1}, \frac{\alpha\pi_2 + \lambda_2 V_2^{N'}}{k_2}\right) \tag{10}$$

Where $V_1^{N'} = \frac{\partial V_1^N}{\partial R}$, $V_2^{N'} = \frac{\partial V_2^N}{\partial R}$.

Substituting (10) into (8) and (9), we have the following equation:

$$\rho V_1^N = \left\{\begin{array}{l}\dfrac{\mu_1^2 + \mu_1\lambda_1 V_1^{N'}}{k_1} + \dfrac{\mu_2\alpha\pi_2 + \mu_2\lambda_2 V_2^{N'}}{k_2} + \pi_1(\dfrac{\alpha^2\pi_2 + a\lambda_2 V_2^{N'}}{k_2} + \theta R) - \dfrac{1}{2}\dfrac{(\mu_1 + \lambda_1 V_1^{N'})^2}{k_1}\\[2ex] + V_1^{N'}(\dfrac{\lambda_1\mu_1 + \lambda_1^2 V_1^{N'}}{k_1} + \dfrac{\lambda_2\alpha\pi_2 + \lambda_2^2 V_2^{N'}}{k_2} - \delta R)\end{array}\right\} \tag{11}$$

$$\rho V_2^N = \left\{\pi_2(\dfrac{\alpha^2\pi_2 + \alpha\lambda_2 V_2^{N'}}{k_2} + \theta R) - \dfrac{1}{2}\dfrac{(\alpha\pi_2 + \lambda_2 V_2^{N'})^2}{k_2} + V_2^{N'}(\dfrac{\lambda_1\mu_1 + \lambda_1^2 V_1^{N'}}{k_1} + \dfrac{\lambda_2\alpha\pi_2 + \lambda_2^2 V_2^{N'}}{k_2} - \delta R)\right\} \tag{12}$$

By observing the structural form of Eqs (11) and (12), it can be inferred that the linear optimal benefit function satisfies the solution of HJB equation.

Assume that the form of the optimal linear function of $V_1^N$, $V_2^N$ is: $V_1^N = c_1 R + c_2$, $V_2^N = d_1 R + d_2$, where $c_1, c_2, d_1, d_2$ is constant. Substitute it into (11), (12) we can get:

$$\rho(c_1 R + c_2) = \left\{ \begin{array}{l} \dfrac{\mu_1^2 + \mu_1 \lambda_1 c_1}{k_1} + \dfrac{\mu_2 \alpha \pi_2 + \mu_2 \lambda_2 d_1}{k_2} + \pi_1 \left( \dfrac{\alpha^2 \pi_2 + \alpha \lambda_2 d_1}{k_2} + \theta R \right) - \dfrac{1}{2} \dfrac{(\mu_1 + \lambda_1 c_1)^2}{k_1} \\ + c_1 \left( \dfrac{\lambda_1 \mu_1 + \lambda_1^2 c_1}{k_1} + \dfrac{\lambda_2 \alpha \pi_2 + \lambda_2^2 d_1}{k_2} - \delta R \right) \end{array} \right\} (13)$$

$$\rho(d_1 R + d_2) = \left\{ \pi_2 \left( \dfrac{\alpha^2 \pi_2 + \alpha \lambda_2 d_1}{k_2} + \theta R \right) - \dfrac{1}{2} \dfrac{(\alpha \pi_2 + \lambda_2 d_1)^2}{k_2} + d_1 \left( \dfrac{\lambda_1 \mu_1 + \lambda_1^2 c_1}{k_1} + \dfrac{\lambda_2 \alpha \pi_2 + \lambda_2^2 d_1}{k_2} - \delta R \right) \right\} (14)$$

According to Eqs (13) and (14), the coefficient of the optimal value can be solved as follows:

$$\left\{ \begin{array}{l} c_1 = \dfrac{\pi_1 \theta}{\rho + \delta} \\ c_2 = \dfrac{1}{\rho} \left[ \dfrac{\mu_1^2}{k_1} + \dfrac{\lambda_1 \mu_1 \pi_1 \theta}{k_1(\rho + \delta)} + \dfrac{\mu_2}{k_2} \left( \alpha \pi_2 + \dfrac{\lambda_2 \pi_2 \theta}{\rho + \delta} \right) + \dfrac{\pi_1}{k_2} \left( \alpha^2 \pi_2 + \dfrac{\alpha \lambda_2 \pi_2 \theta}{\rho + \delta} \right) - \dfrac{1}{2k_1} \left( \mu_1 + \dfrac{\lambda_1 \pi_1 \theta}{\rho + \delta} \right)^2 + \\ \dfrac{\pi_1 \theta}{\rho + \delta} \left( \dfrac{\lambda_1 \mu_1}{k_1} + \dfrac{\lambda_1^2 \pi_1 \theta}{k_1(\rho + \delta)} + \dfrac{\lambda_2 \alpha \pi_2}{k_2} + \dfrac{\lambda_2^2 \pi_2 \theta}{k_2(\rho + \delta)} \right) \right] \end{array} \right. (15)$$

$$\left\{ \begin{array}{l} d_1 = \dfrac{\pi_2 \theta}{\rho + \delta} \\ d_2 = \dfrac{1}{\rho} \left[ \left( \dfrac{\pi_2^2 \alpha^2}{k_2} + \dfrac{\pi_2 \alpha \lambda_2 \theta}{k_2(\rho + \delta)} \right) - \dfrac{1}{2k_2} \left( \alpha \pi_2 + \dfrac{\lambda_2 \pi_2 \theta}{\rho + \delta} \right)^2 + \dfrac{\pi_2 \theta}{\rho + \delta} \left( \dfrac{\lambda_1 \mu_1}{k_1} + \right. \right. \\ \left. \left. \dfrac{\lambda_1^2 \pi_1 \theta}{k_1(\rho + \delta)} + \dfrac{\lambda_2 \alpha \pi_2}{k_2} + \dfrac{\lambda_2^2 \pi_2 \theta}{k_2(\rho + \delta)} \right) \right] \end{array} \right. (16)$$

Substituting $c_1, c_2, d_1, d_2$ into the linear function, the optimal benefit function can be obtained as follows:

$$\begin{array}{l} V_1^{N*} = \dfrac{\pi_1 \theta}{\rho + \delta} R + \dfrac{1}{\rho} \left[ \dfrac{\mu_1^2}{k_1} + \dfrac{\lambda_1 \mu_1 \pi_1 \theta}{k_1(\rho + \delta)} + \dfrac{\mu_2}{k_2} \left( \alpha \pi_2 + \dfrac{\lambda_2 \pi_2 \theta}{\rho + \delta} \right) + \dfrac{\pi_1}{k_2} \left( \alpha^2 \pi_2 + \dfrac{\alpha \lambda_2 \pi_2 \theta}{\rho + \delta} \right) - \\ \dfrac{1}{2k_1} \left( \mu_1 + \dfrac{\lambda_1 \pi_1 \theta}{\rho + \delta} \right)^2 + \dfrac{\pi_1 \theta}{\rho + \delta} \left( \dfrac{\lambda_1 \mu_1}{k_1} + \dfrac{\lambda_1^2 \pi_1 \theta}{k_1(\rho + \delta)} + \dfrac{\lambda_2 \alpha \pi_2}{k_2} + \dfrac{\lambda_2^2 \pi_2 \theta}{k_2(\rho + \delta)} \right) \right] \end{array} (17)$$

$$\begin{array}{l} V_2^{N*} = \dfrac{\pi_2 \theta}{\rho + \delta} R + \dfrac{1}{\rho} \left[ \left( \dfrac{\pi_2^2 \alpha^2}{k_2} + \dfrac{\pi_2 \alpha \lambda_2 \theta}{k_2(\rho + \delta)} \right) - \dfrac{1}{2k_2} \left( \alpha \pi_2 + \dfrac{\lambda_2 \pi_2 \theta}{\rho + \delta} \right)^2 + \\ \dfrac{\pi_2 \theta}{\rho + \delta} \left( \dfrac{\lambda_1 \mu_1}{k_1} + \dfrac{\lambda_1^2 \pi_1 \theta}{k_1(\rho + \delta)} + \dfrac{\lambda_2 \alpha \pi_2}{k_2} + \dfrac{\lambda_2^2 \pi_2 \theta}{k_2(\rho + \delta)} \right) \right] \end{array} (18)$$

Substituting (17) and (18) into (10), we can get (7), prove that the end.

**Corollary 1** In the compensation mechanism without cost sharing, the optimal ecological compensation level of government departments $S_1$ is negatively correlated with the discount rate $\rho$, the coefficient of compensation cost $k_1$ and the coefficient of natural decline of corporate reputation $\delta$, and is positively correlated with the coefficient of government compensation $\lambda_1$, the coefficient of government benefit $\mu_1$ and the coefficient of consumer demand $\theta$.

**Corollary 2** In a compensation mechanism without cost sharing, the optimal ecological compensation level $S_2$ is negatively correlated with the discount rate $\rho$, the compensation cost

coefficient $k_2$ and the natural decline coefficient $\delta$ of the enterprise reputation, and is positively correlated with advertising effect coefficient $\alpha$, enterprise revenue coefficient $\pi_2$, enterprise compensation input coefficient $\lambda_2$ and corporate social reputation coefficient $\theta$.

In conclusion, the optimal compensation level of both the government and the enterprise is positively correlated with its earnings, which indicates that both the government and the enterprise make decisions based on the principle of maximizing their own benefits, without considering the overall benefits of the system. The ecological compensation level of an enterprise is affected by its compensation cost coefficient. When the compensation cost coefficient of an enterprise decreases, the enterprise will increase the investment level of ecological compensation.

## 4.2 Compensation cost sharing mechanism

Under the compensation cost sharing mechanism. In order to encourage enterprise to actively conduct ecological compensation, government provide subsidies for the compensation cost of the enterprise. Both government and enterprise make decisions based on maximizing returns. The government first makes decisions about the level of ecological compensation and the proportion of cost sharing, and then enterprise make decisions based on the government's decisions.

$$\max_{S_1(t),L} J_1^D = \int_0^\infty e^{-\rho t}[\mu_1 S_1(t) + \mu_2 S_2(t) + \pi_1 Q(t) - C_1(t) - L(t)C_2(t)]dt \tag{19}$$

$$\max_{S_2(t)} J_2^D = \int_0^\infty e^{-\rho t}[\pi_2 Q(t) - (1 - L(t))C_2(t)]dt \tag{20}$$

**Proposition 2** Under the mechanism of the government sharing the compensation cost of enterprise, the dynamic feedback balance between government and enterprise on the level of ecological compensation and the proportion of cost sharing are:

$$S_1^{D*} = \frac{\mu_1(\rho + \delta) + \lambda_1 \pi_1 \theta}{k_1(\rho + \delta)} \tag{21}$$

$$S_2^{D*} = \begin{cases} \dfrac{[2(\mu_2 + \pi_1\alpha) + \pi_2\alpha](\rho + \delta) + \lambda_2\theta(\alpha\pi_2 + 2\pi_1)}{2(\rho + \delta)k_2}, & 2A - B > 0 \\[3mm] \dfrac{\alpha\pi_2(\rho + \delta) + \lambda_2\pi_2\theta}{k_2(\rho + \delta)}, & 2A - B < 0 \end{cases} \tag{22}$$

$$L^* = \begin{cases} \dfrac{2(\mu_2 + \pi_1\alpha + \frac{\lambda_2\pi_1\theta}{\rho + \delta}) - (\pi_2\alpha + \frac{\lambda_2\pi_2\theta}{\rho + \delta})}{2(\mu_2 + \pi_1\alpha + \frac{\lambda_2\pi_1\theta}{\rho + \delta}) + (\pi_2\alpha + \frac{\lambda_2\pi_2\theta}{\rho + \delta})}, & 2A - B > 0 \\[3mm] 0, & 2A - B < 0 \end{cases} \tag{23}$$

Where $A = \mu_2 + \pi_1\alpha + \frac{\pi_1\theta}{\rho+\delta}\lambda_2$, $B = \pi_2\alpha + \frac{\pi_2\theta}{\rho+\delta}\lambda_2$

**Proof** Using reverse induction method to solve Stackelberg game, the optimal value function $V_2^D$ of the enterprise satisfies the HJB equation, as shown below:

$$\rho V_2^D = \max_{S_2} \left\{ \pi_2(\alpha S_2 + \theta R) - (1 - L)\frac{k_2 S_2^2}{2} + V_2^{D'}(\lambda_1 S_1 + \lambda_2 S_2 - \delta R) \right\} \tag{24}$$

In Eq (22), the first-order condition of enterprise compensation level $S_2$ is:

$$S_2^D = \frac{\pi_2\alpha + \lambda_2 V_2^{D'}}{(1 - L)k_2} \tag{25}$$

Considering that enterprise will decide countermeasure according to the government's decision, the government will decide its decision according to the rational response of enterprise. At this time, we can get the HJB equation of government departments as follows:

$$\rho V_1^D = \max_{S_1, L} \left\{ \begin{array}{l} \mu_1 S_1 + \dfrac{\mu_2(\pi_2\alpha + \lambda_2 V_2^{D'})}{(1-L)k_2} + \pi_1(\alpha S_2 + \theta R) - \dfrac{k_1 S_1^2}{2} - \dfrac{L(\pi_2\alpha + \lambda_2 V_2^{D'})^2}{2k_2(1-L)^2} \\[3mm] + V_1^{D'}\left(\lambda_1 S_1 + \dfrac{\lambda_2(\pi_2\alpha + \lambda_2 V_2^{D'})}{(1-L)k_2} - \delta R\right) \end{array} \right\} \tag{26}$$

To solve the first-order conditions of government compensation level $S_1$ and government cost-sharing proportion $L$ respectively, we can get:

$$S_1 = \frac{\mu_1 + \lambda_1 V_1^{D'}}{k_1} \tag{27}$$

$$L = \begin{cases} \dfrac{2(\mu_2 + \pi_1\alpha + V_1^{D'}\lambda_2) - (\pi_2\alpha + V_2^{D'}\lambda_2)}{2(\mu_2 + \pi_1\alpha + V_1^{D'}\lambda_2) + (\pi_2\alpha + V_2^{D'}\lambda_2)}, & 2A - B > 0 \\[3mm] 0, & 2A - B < 0 \end{cases} \tag{28}$$

Where $V_1^{D'} = \frac{\partial V_1^D}{\partial R}$, $V_2^{D'} = \frac{\partial V_2^D}{\partial R}$; $A = \mu_2 + \pi_1\alpha + \lambda_2 V_1^{D'}$, $B = \pi_2\alpha + \lambda_2 V_2^{D'}$

Substitute (25), (27) and (28) into (24) and (26), and we can get:

$$\rho V_2^D = \max_{S_2} \left\{ \pi_2\left(\frac{\alpha(2A+B)B}{2Bk_2} + \theta R\right) - \frac{B(2A+B)}{4k_2} + V_2^{D'}\left(\frac{\lambda_1\mu_1 + \lambda_1^2 V_1^{D'}}{k_1} + \frac{\lambda_2(2A+B)}{2k_2} - \delta R\right) \right\} \tag{29}$$

$$\rho V_1^D = \max_{S_1, L} \left\{ \begin{array}{l} \dfrac{\mu_1^2 + \mu_1\lambda_1 V_1^{D'}}{k_1} + \dfrac{\mu_2(2A+B)}{2k_2} + \pi_1\left(\dfrac{\alpha(2A+B)}{2k_2} + \theta R\right) - \dfrac{(\mu_1 + \lambda_1 p_1)^2}{2k_1} - \dfrac{(2A-B)(2A+B)}{8k_2} \\[3mm] + V_1^{D'}\left(\dfrac{\lambda_1\mu_1 + \lambda_1^2 p_1}{k_1} + \dfrac{\lambda_2(2A+B)}{2k_2} - \delta R\right) \end{array} \right\} \tag{30}$$

Assume that the optimal value function satisfying the HJB equation is:

$$V_1^D = p_1 R + p_2, \quad V_2^D = q_1 R + q_2 \tag{31}$$

Where $p_1$, $p_2$ and $q_1$, $q_2$ are constants. By substituting Eq (31) into Eqs (29) and (30) respectively, we have:

$$\rho(p_1 R + p_2) = \left\{ \begin{array}{l} \dfrac{\mu_1^2 + \mu_1 \lambda_1 p_1}{k_1} + \dfrac{\mu_2(2A+B)}{2k_2} + \pi_1(\dfrac{\alpha(2A+B)}{2k_2} + \theta R) - \dfrac{(\mu_1 + \lambda_1 p_1)^2}{2k_1} - \dfrac{(2A-B)(2A+B)}{8k_2} \\[3mm] + p_1(\dfrac{\lambda_1 \mu_1 + \lambda_1^2 p_1}{k_1} + \dfrac{\lambda_2(2A+B)}{2k_2} - \delta R) \end{array} \right\} (32)$$

$$\rho(q_1 R + q_2) = \left\{ \pi_2(\dfrac{\alpha(2A+B)B}{2Bk_2} + \theta R) - \dfrac{B(2A+B)}{4k_2} + q_1(\dfrac{\lambda_1 \mu_1 + \lambda_1^2 p_1}{k_1} + \dfrac{\lambda_2(2A+B)}{2k_2} - \delta R) \right\} (33)$$

By sorting out the function equations, the coefficient of the optimal linear function can be obtained as follows:

$$\left\{ \begin{array}{l} p_1 = \dfrac{\pi_1 \theta}{\rho + \delta} \\[3mm] p_2 = \dfrac{1}{\rho} [\dfrac{\mu_1^2}{k_1} + \dfrac{\mu_1 \lambda_1 \pi_1 \theta}{k_1(\rho+\delta)} + \dfrac{\mu_2(2A+B)}{2k_2} + \pi_1(\dfrac{\alpha(2A+B)}{2k_2}) - \dfrac{1}{2k_1}(\mu_1 + \dfrac{\lambda_1 \pi_1 \theta}{\rho+\delta})^2 - \\[3mm] \dfrac{(2A-B)(2A+B)}{8k_2} + \dfrac{\pi_2 \theta}{\rho+\delta}(\dfrac{\lambda_1 \mu_1}{k_1} + \dfrac{\lambda_1^2 \pi_1 \theta}{k_1(\rho+\delta)} + \dfrac{\lambda_2(2A+B)}{2k_2})] \end{array} \right. (34)$$

$$\left\{ \begin{array}{l} q_1 = \dfrac{\pi_2 \theta}{\rho + \delta} \\[3mm] q_2 = \dfrac{1}{\rho}[\pi_2(\dfrac{\alpha(2A+B)B}{2Bk_2}) - \dfrac{B(2A+B)}{4k_2} + \dfrac{\pi_2 \theta}{\rho+\delta}(\dfrac{\lambda_1 \mu_1}{k_1} + \dfrac{\lambda_1^2 \pi_1 \theta}{k_1(\rho+\delta)} + \dfrac{\lambda_2(2A+B)}{2k_2})] \end{array} \right. (35)$$

Where $A = \mu_2 + \pi_1 \alpha + \frac{\pi_1 \theta}{\rho+\delta} \lambda_2$, $B = \pi_2 \alpha + \lambda_2 \frac{\pi_2 \theta}{\rho+\delta}$

Substitute $p_1$, $p_2$, $q_1$, $q_2$ into the optimal value function formula of government and enterprise, and we can get:

$$V_1^{D*} = \dfrac{\pi_1 \theta}{\rho+\delta} R + \dfrac{1}{\rho}[\dfrac{\mu_1^2}{k_1} + \dfrac{\mu_1 \lambda_1 \pi_1 \theta}{k_1(\rho+\delta)} + \dfrac{\mu_2(2A+B)}{2k_2} + \pi_1(\dfrac{\alpha(2A+B)}{2k_2}) - \dfrac{1}{2k_1}(\mu_1 + \dfrac{\lambda_1 \pi_1 \theta}{\rho+\delta})^2 - \dfrac{(2A-B)(2A+B)}{8k_2} + \dfrac{\pi_2 \theta}{\rho+\delta}(\dfrac{\lambda_1 \mu_1}{k_1} + \dfrac{\lambda_1^2 \pi_1 \theta}{k_1(\rho+\delta)} + \dfrac{\lambda_2(2A+B)}{2k_2})] \quad (36)$$

$$V_2^{D*} = \dfrac{\pi_2 \theta}{\rho+\delta} R + \dfrac{1}{\rho}[\pi_2(\dfrac{\alpha(2A+B)B}{2Bk_2}) - \dfrac{B(2A+B)}{4k_2} + \dfrac{\pi_2 \theta}{\rho+\delta}(\dfrac{\lambda_1 \mu_1}{k_1} + \dfrac{\lambda_1^2 \pi_1 \theta}{k_1(\rho+\delta)} + \dfrac{\lambda_2(2A+B)}{2k_2})] (37)$$

Substituting the derivatives of (36) and (37) into (25), (27) and (28), we can get (21), (22) and (23).

**Corollary 3** Only in the case of $2A>B$, government will choose to share the ecological compensation cost of enterprise. The sharing proportion $L$ of government is negatively correlated with coefficient $\pi_2$.

**Corollary 4** In the case of government sharing of enterprise compensation cost, enterprise's ecological compensation level is greater than that in the case of no sharing. Enterprise's ecological compensation level $S_2(t)$ is positively correlated with coefficient $\mu_2$, coefficient $\pi_1$ and coefficient $\pi_2$.

## 4.3 Collaborative cooperation mechanism

In order to improve the social benefits of government and improve the social reputation and income of enterprise, both government and enterprise have the motivation to carry out ecological compensation.

This part discusses the cooperative compensation mechanism between the government and enterprise. The government and the enterprise determine the optimal compensation level with the goal of maximizing the system benefit, so as to achieve the optimal benefit of the system.

**Proposition 3** In the case of government-enterprise coordination compensation, the optimal compensation levels of both parties are as follows:

$$S_1^{C*} = \frac{\mu_1(\rho + \delta) + \lambda_1(\pi_1 + \pi_2)\theta}{k_1(\rho + \delta)} \tag{38}$$

$$S_2^{C*} = \frac{(\rho + \delta)[\mu_2 + (\pi_1 + \pi_2)\alpha] + \lambda_2(\pi_1 + \pi_2)\theta}{k_2(\rho + \delta)} \tag{39}$$

**Prove** When the government and enterprise cooperate to carry out ecological compensation, the two sides aim at maximizing the system benefits, and the objective function of the system benefits is:

$$\max_{S_1,S_2} J_3^C = \int_0^\infty e^{-\rho t}[\mu_1 S_1 + \mu_2 S_2 + (\pi_1 + \pi_2)Q - C_1 - C_2]dt \tag{40}$$

At this point, the overall optimal benefit function $V_S^C$ of the system satisfies the HJB equation, as shown below:

$$\rho V_3^C = \max_{S_1,S_2}\left\{\mu_1 S_1 + \mu_2 S_2 + (\pi_1 + \pi_2)(\alpha S_2 + \theta R) - \frac{1}{2}k_1 S_1^2 - \frac{1}{2}k_2 S_2^2 + V_3^{C'}(\lambda_1 S_1 + \lambda_2 S_2 - \delta R)\right\} \tag{41}$$

By solving the first-order condition of $S_1$, $S_2$, we can get:

$$S_1 = \frac{\mu_1 + \lambda_1 V_3^{C'}}{k_1} \tag{42}$$

$$S_2 = \frac{\mu_2 + (\pi_1 + \pi_2)\alpha + \lambda_2 V_3^{C'}}{k_2} \tag{43}$$

Where $V_3^{C'} = \frac{\partial V_3^C}{\partial R}$, substituting (42), (43) into (41), we can get:

$$\rho V_3^C = \left\{\begin{array}{l}\dfrac{\mu_1^2 + \mu_1\lambda_1 V_3^{C'}}{k_1} + \dfrac{\mu_2^2 + \mu_2(\pi_1 + \pi_2)\alpha + \mu_2\lambda_2 V_3^{C'}}{k_2} + (\pi_1 + \pi_2)(\dfrac{\alpha\mu_2 + (\pi_1 + \pi_2)\alpha^2 + \alpha\lambda_2 V_3^{C'}}{k_2} + \\[2mm] \theta R) - \dfrac{1}{2}\dfrac{(\mu_1 - \lambda_1 V_3^{C'})^2}{k_1} - \dfrac{1}{2}\dfrac{(\mu_2 + (\pi_1 + \pi_2)\alpha + \lambda_2 V_3^{C'})^2}{k_2} + V_3^{C'}(\dfrac{\lambda_1\mu_1 - \lambda_1^2 V_3^{C'}}{k_1} + \\[2mm] \dfrac{\lambda_2\mu_2 + \lambda_2(\pi_2 + \pi_2)\alpha + \lambda_2^2 V_3^{C'}}{k_2} - \delta R)\end{array}\right\} \tag{44}$$

Similarly, the linearly optimal benefit function of $R$ is the solution of HJB equation, and let

$$V_3^C(R) = m_1 R + m_2 \tag{45}$$

Where $m_1$, $m_2$ is a constant. Substituting (45) into (44), the coefficient of the optimal benefit function can be obtained as follows:

$$\begin{cases} m_1 = \dfrac{(\pi_1 + \pi_2)\theta}{\rho + \delta} \\[2ex] m_2 = \dfrac{1}{\rho}[\dfrac{\mu_1^2 - \mu_1\lambda_1 m_1}{k_1} + \dfrac{\mu_2^2 + \mu_2(\pi_1 + \pi_2)\alpha + \mu_2\lambda_2 m_1}{k_2} + (\pi_1 + \pi_2)(\dfrac{\alpha\mu_2 + (\pi_1 + \pi_2)\alpha^2 + \alpha\lambda_2 m_1}{k_2}) - \\[2ex] \dfrac{1}{2}\dfrac{(\mu_1 - \lambda_1 m_1)^2}{k_1} - \dfrac{1}{2}\dfrac{(\mu_2 + (\pi_1 + \pi_2)\alpha + \lambda_2 m_1)^2}{k_2} + m_1(\dfrac{\lambda_1\mu_1 - \lambda_1^2 m_1}{k_1} + \dfrac{\lambda_2\mu_2 + \lambda_2(\pi_1 + \pi_2)\alpha + \lambda_2^2 m_1}{k_2})] \end{cases} \quad (46)$$

Substituting $m_1$, $m_2$ into Eq (45), the optimal benefit function of the system can be obtained as follows:

$$V_3^{C*} = \dfrac{(\pi_1 + \pi_2)\theta}{\rho + \delta}R + \dfrac{1}{\rho}[\dfrac{\mu_1^2 - \mu_1\lambda_1 m_1}{k_1} + \dfrac{\mu_2^2 + \mu_2(\pi_1 + \pi_2)\alpha + \mu_2\lambda_2 m_1}{k_2} + (\pi_1 + \pi_2)(\dfrac{\alpha\mu_2 + (\pi_1 + \pi_2)\alpha^2 + \alpha\lambda_2 m_1}{k_2})$$
$$-\dfrac{1}{2}\dfrac{(\mu_1 - \lambda_1 m_1)^2}{k_1} - \dfrac{1}{2}\dfrac{(\mu_2 + (\pi_1 + \pi_2)\alpha + \lambda_2 m_1)^2}{k_2} + m_1(\dfrac{\lambda_1\mu_1 - \lambda_1^2 m_1}{k_1} + \dfrac{\lambda_2\mu_2 + \lambda_2(\pi_1 + \pi_2)\alpha + \lambda_2^2 m_1}{k_2})] \quad (47)$$

Substituting (47) into (42) and (43), the optimal compensation input level of the government and enterprise can be obtained as (38) and (39).

**Corollary 5** In the process of government-enterprise cooperative compensation, the optimal compensation level $S_1^{C*}$ of the government, the optimal compensation level $S_2^{C*}$ of the enterprise and the overall income $V_3^{C*}$ of the system are all positively correlated with the sum of coefficients $(\pi_1 + \pi_2)$.

## 5 Comparative analysis

According to the solution results of the above model, the optimal compensation level and benefits of government and enterprise under no cost sharing mechanism, cost sharing mechanism and collaborative mechanism are compared and analyzed, and the following conclusions can be drawn:

**Proposition 4** In the collaborative cooperation mechanism, the ecological compensation level of government and enterprise reaches the highest level.

**Prove** For the government, according to Eqs (7), (21) and (38), we can get:

$$S_1^{N*} = S_1^{D*} = \dfrac{\mu_1(\rho + \delta) + \lambda_1\pi_1\theta}{k_1(\rho + \delta)}, S_1^{C*} = \dfrac{\mu_1(\rho + \delta) + \lambda_1(\pi_1 + \pi_2)\theta}{k_1(\rho + \delta)}$$

Therefore, $S_1^{N*} = S_1^{D*} < S_1^{C*}$.

For enterprises, we can be concluded from Eqs (7), (22) and (39) that, when $2A > B$, we have:

$$S_2^{D*} - S_2^{N*} = \dfrac{(2\mu_2 + 2\pi_1\alpha)(\rho + \delta) + 2\lambda_2\pi_1\theta - \pi_2\alpha(\rho + \delta) - \lambda_2\pi_2\theta}{2(\rho + \delta)k_2} = \dfrac{2A - B}{2k_2} > 0 \quad (48)$$

$$S_2^{C*} - S_2^{D*} = \dfrac{(\rho + \delta)\pi_2\alpha + \lambda_2\pi_2\theta}{2k_2(\rho + \delta)} > 0 \quad (49)$$

So, when $2A > B$, we get $S_2^{C*} > S_2^{D*} > S_2^{N*}$.

As can be seen from Proposition 4, when $2A > B$, compared with no cost-sharing mechanism, the government's ecological compensation level remains unchanged under the cost-

sharing mechanism, while the enterprise's ecological compensation level increases. Under the cooperative compensation mechanism, the government and the enterprise have the highest ecological compensation level.

**Proposition 5** Compared with no cost-sharing mechanism, under the cost-sharing mechanism, both the government and the enterprise can achieve Pareto improvement in revenue.

**Prove** For the government, according to Eqs (17) and (36), we can get:

$$V_1^{D*} - V_1^{N*} = \frac{\left[(2\mu_2 + 2\pi_1\alpha - \pi_2\alpha)(\rho + \delta) + \lambda_2\theta(2\pi_1 - \pi_2)\right]^2}{8k_2\rho(\rho + \delta)^2} > 0 \tag{50}$$

For enterprise, according to (18) and (37), we can calculate:

$$V_2^{D*} - V_2^{N*} = \frac{\left[(2\mu_2 + 2\pi_1\alpha - \pi_2\alpha)(\rho + \delta) + \lambda_2\theta(2\pi_1 - \pi_2)\right]^2}{4k_2\rho} > 0 \tag{51}$$

It can be seen from Proposition 5, the benefits of both the government and enterprise in the cost-sharing mechanism are greater than the benefits without cost-sharing. This indicates that in the cost-sharing mechanism, the Pareto improvement of the system is realized through the government sharing a certain proportion of the ecological compensation cost of the enterprise, and both government and corporate profits have improved.

**Corollary 6** The Pareto improvement effect of cost sharing mechanism on government and enterprise's revenue is negatively correlated with coefficient $k_2$, and positively correlated with coefficient $\mu_2$ and coefficient difference $(2\pi_1 - \pi_2)$.

**Proposition 6** Under the government-enterprise cooperative compensation mechanism, the optimal benefit of the system is greater than that under the other two mechanisms.

**Proof** According to proposition 5, it can be known that:

$$V_1^{D*} + V_2^{D*} > V_1^{N*} + V_2^{N*} \tag{52}$$

According to (18), (37) and (47), We have that:

$$V_3^{C*} - (V_1^{D*} + V_2^{D*}) > 0 \tag{53}$$

we can get $V_3^{C*} > V_1^{D*} + V_2^{D*} > V_1^{N*} + V_2^{N*}$.

In conclusion, when $2A > B$, compared with no cost sharing mechanism, the cost sharing mechanism can achieve Pareto improvement of the system. The optimal return of the system under the cost-sharing mechanism is greater than that without the cost-sharing mechanism, and the optimal return of the system under the collaborative mechanism is greater than that under the cost-sharing mechanism. However, it is worth noting that the government and enterprise will choose the cooperative compensation strategy only when the benefits of their respective cooperative compensation are greater than the benefits of non-cooperative compensation. As for the proportion of the government and the enterprise in the system benefits, it depends on the negotiation ability of the government and enterprise. If the final income distribution scheme is reasonable and feasible, then the two parties will get the optimal benefits under the cooperative compensation mechanism.

## 6 Analysis of calculation examples

In order to verify the effectiveness of the basin ecological compensation model established in this paper. We selected the cross-provincial ecological compensation pilot project of Xinanjiang River Basin as an example for numerical simulation analysis. Since the ecological compensation mechanism proposed in this paper is a new theory, direct real data cannot be

obtained now. Therefore, this paper referred to the existing literature [27, 28] and combined with the relevant data in China Environmental Statistics Yearbook and Local statistical yearbook, make the parameter setting as practical as possible. The parameters in the paper are determined as follows:

$$\rho = 0.2, \theta = 0.7, \delta = 0.7, \mu_1 = 1.2, \mu_2 = 1, \pi_1 = 1, \pi_2 = 1.5, \alpha = 1, \lambda_1 = 1, \lambda_2 = 2, k_1 = 5, k_2 = 6,$$

the following numerical analysis can be obtained.

1. The improvement effect of the cost-sharing mechanism relative to that of no cost-sharing mechanism is analyzed in Fig 3. We can be seen from Fig 3, the cost-sharing mechanism can achieve Pareto improvement of government and enterprise earnings, and the improvement effect of enterprise earnings is better than that of government earnings. Since the government shares part of the compensation cost of enterprises, enterprises will make greater efforts to make ecological compensation to improve their advertising effect and social reputation. This will increase the demand of consumers and affect the revenue of government and enterprises, so both parties can achieve Pareto improvement.

2. In Fig 4, we simulate the effect of parameter changes on the cost-sharing ratio. X-axis represents the change in parameters, and Y-axis represents the government's share of compensation costs; $\mu_2$ represents the enterprise compensation benefit coefficient; $\pi_1$ represents the government demand income coefficient and $\pi_2$ represents the enterprise demand-income coefficient. We can be seen from Fig 4, as the influence coefficient $\mu_2$ and $\pi_1$ increases, the proportion of government's sharing of compensation costs to enterprises increases, and the growth rate of sharing ratio slows down gradually. The value of coefficient $\mu_2$ can be used as an indicator to measure the degree of ecological environmental damage. When $\mu_2$ is large, it indicates that the ecological damage is more serious and the government has a greater demand for direct compensation from enterprise. In order to encourage enterprise to compensate actively, so the government is willing to share a larger proportion of the compensation costs of enterprise. When the coefficient $\pi_2$ increases, in order to obtain profits the

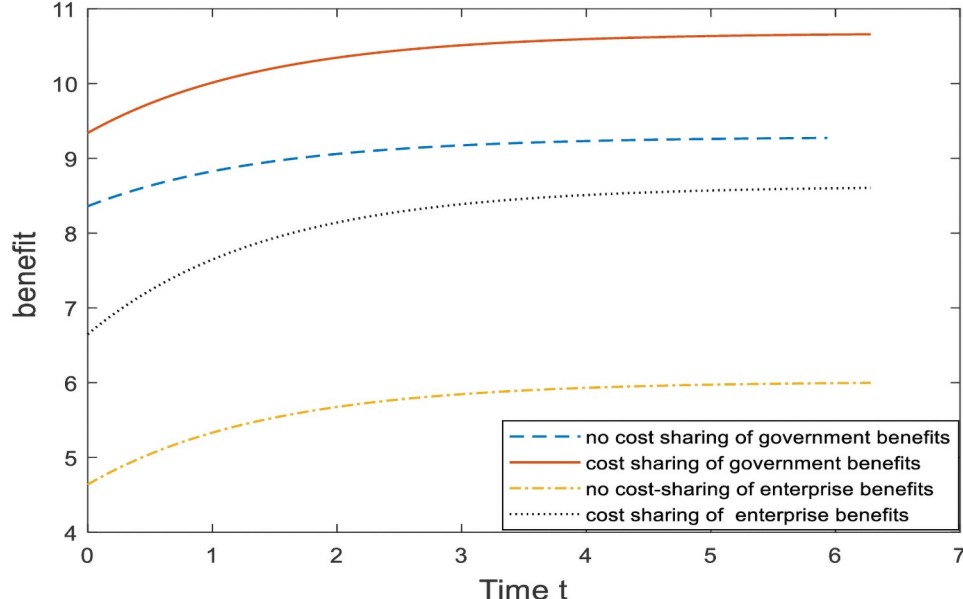

**Fig 3. Comparison of government and enterprise revenue.**

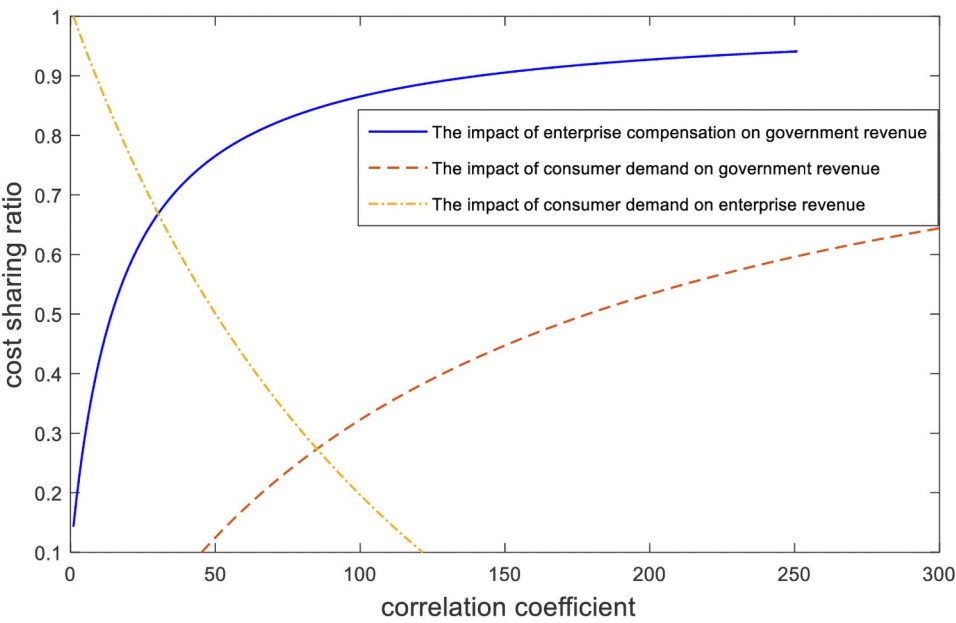

**Fig 4. The influence of parameters on the proportion of sharing.**

enterprise will actively participate in ecological compensation, while the government will reduce the compensation ratio.

3. In Fig 5, we simulate the influence of parameters on the Pareto improvement effect of the cost-sharing mechanism. X axis represents the cost compensation coefficient of the enterprise, and Y axis represents the income improvement level of government and enterprise; $k_2$ represents cost coefficients of enterprise. The red line and the blue line respectively

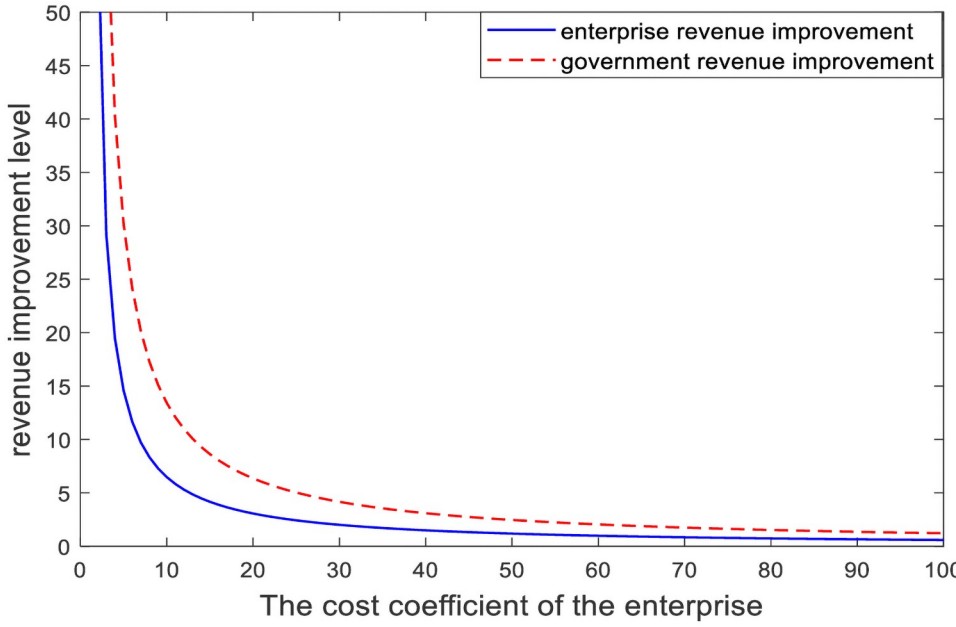

**Fig 5. Influence of parameters on cost sharing improvement effect.**

represent the revenue change of government and enterprise caused by the change of coefficient $k_2$. As we can see from Fig 5, the improvement effect increases with the reduction of the cost coefficient $k_2$ of ecological compensation. The larger the enterprise's ecological compensation cost coefficient is, the more the enterprise's ecological compensation cost is and the less obvious the effect of the government's compensation cost sharing is. Therefore, the government should also consider the compensation cost of enterprise when making decisions and give as much support as possible to the compensation cost of enterprise.

4. Fig 6 compares the total benefits of the system under the three compensation mechanisms. In the graph, the X axis represents the passage of time, and the Y axis represents the change in the total revenue of the system. The three curves represent the non-compensation mechanism respectively. As can be seen from Fig 6, under the government-enterprise cooperation mechanism, the overall benefits of the system are the largest; in the absence of cost sharing mechanism, the system benefits are the least; in the case of cost sharing mechanism, the overall revenue of the system realizes Pareto improvement. Under the government-enterprise cooperative cooperation mechanism, the total benefit of the system is far greater than that under the other two mechanisms, which fully indicates that the cooperative decision is superior to the non-cooperative decision.

Table 2 is the sensitivity analysis of the influence of parameter changes on equilibrium results, where "+" represents increase, "-" represents decrease, and "o" represents constant. As can be seen from Table 2, the influences of different parameter changes on the equilibrium results are as follows:

1. The influence coefficient $\theta$ of corporate reputation on the demand function has a positive effect on all variables; the discount rate $\rho$ and the attenuation coefficient $\delta$ of corporate reputation have a negative effect on all variables; in addition, the ecological compensation cost coefficient $k_1$, $k_2$ of government and enterprise has a negative effect no effect on all variables in the table.

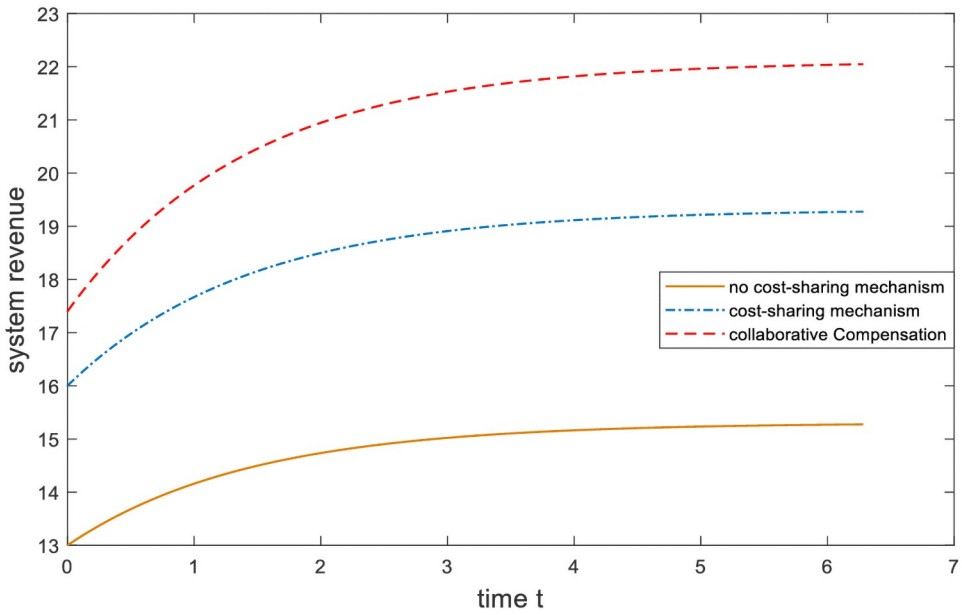

**Fig 6. Comparison of system benefits under different mechanisms.**

**Table 2. Sensitivity analysis of relevant parameters.**

| Parameter | $V_1^{N*}$ | $V_2^{N*}$ | $S_1^{N*}$ | $S_2^{N*}$ | $V_1^{D*}$ | $V_2^{D*}$ | $S_1^{D*}$ | $S_2^{D*}$ | $L^*$ | $V_3^{C*}$ | $S_1^{C*}$ | $S_2^{C*}$ |
|---|---|---|---|---|---|---|---|---|---|---|---|---|
| $\theta = 0.5 \rightarrow 2.1$ | + | + | + | + | + | + | + | + | + | + | + | + |
| $\delta = 0.1 \rightarrow 0.8$ | - | - | - | - | - | - | - | - | - | - | - | - |
| $\rho = 0.1 \rightarrow 0.9$ | - | - | - | - | - | - | - | - | - | - | - | - |
| $k_1 = 3.5 \rightarrow 12$ | - | - | - | o | - | - | - | o | - | - | - | o |
| $k_2 = 4.5 \rightarrow 16$ | - | - | o | - | - | - | o | - | - | - | o | - |
| $\mu_1 = 0.8 \rightarrow 2.4$ | + | + | + | o | + | + | + | o | o | + | + | o |
| $\mu_2 = 0.5 \rightarrow 1.5$ | + | o | o | o | + | + | o | + | + | + | o | + |
| $\pi_1 = 0.6 \rightarrow 1.5$ | + | + | + | o | + | + | + | o | + | + | + | + |
| $\pi_2 = 1.6 \rightarrow 4.8$ | + | + | o | + | + | + | o | + | - | + | + | + |
| $\alpha = 0.5 \rightarrow 3.5$ | + | + | o | + | + | + | o | + | - | + | o | + |
| $\lambda_1 = 0.8 \rightarrow 3.2$ | + | + | + | o | + | + | + | o | o | + | + | o |
| $\lambda_2 = 1.5 \rightarrow 4.5$ | + | + | o | + | + | + | o | + | - | + | o | + |

2. Coefficient $\mu_1$ and coefficient $\lambda_1$ have the same influence on the variation trend of variables in the table, they are positively correlated with government revenue, enterprise revenue, compensation ratio and government compensation level under the three compensation mechanisms; they are positively correlated with the total income under the cooperation mechanism; however, they are no effect on the compensation level and cost-sharing ratio of enterprises.

3. The influence coefficient $\mu_2$ is positively correlated with the government revenue, enterprise revenue and enterprise compensation level under the three compensation mechanisms; it is positively correlated with the total revenue under the cooperation mechanisms; it is independent of the level of government compensation; it is positively correlated with the proportion of government cost sharing.

4. Coefficient $\alpha$ and coefficient $\lambda_2$ also have the same influence on the variation trend of variables in the table, they are positively correlated with the government income, the enterprise income, the enterprise compensation level and the total cooperation income; they are negatively correlated with the cost-sharing ratio; they are no effect on the level of government compensation.

## 7 Conclusions and recommendations

This paper studies the basin ecological compensation system which is dominated by government and participated by enterprise. Firstly, it is assumed that the ecological compensation behavior of the government and enterprise will improve the social reputation of enterprise, and the ecological compensation behavior of enterprise will produce advertising effect in the short term, and the consumer demand is affected by both advertising effect and social reputation. Then, differential game theory is used to compare the effects of no-cost sharing, cost-sharing and cooperative cooperation mechanisms on the revenue of government and enterprise. The following conclusions can be drawn by solving the model:

Firstly, the cost-sharing mechanism can achieve the Pareto improvement of government and enterprise earnings, but the cost-sharing mechanism is conditional (2B>A). This paper explains that enterprise tend to take the initiative of ecological compensation when the ecological environment serious pollution. It also verifies the rationality of subsidies and preferential tax policies for ecological compensation enterprise in China.

Secondly, when the government and enterprise transition from no cost-sharing mechanism to cost-sharing mechanism, the compensation levels of the government and enterprise will change differently. Enterprise will increase their own compensation levels because the government shares a certain proportion of the compensation costs, while the government's compensation level will remain unchanged.

Finally, when the government and enterprise cooperate, the benefits of both parties and the overall benefits of the system will reach the optimal. It also provides a reference for the government-enterprise cooperation and cooperative compensation.

In this paper, the differential game model is used to study the ecological compensation problem, which gets rid of the original research model from the qualitative perspective. Considering the influence of enterprise reputation on customers demand, we take the enterprise sales as the decision variable. The dynamic process of decision-making between government and enterprise in the process of ecological compensation can be described more accurately. This study has important theoretical and practical significance. The theoretical significance of this paper breaks away from the routine of previous scholars who studied the game between government and enterprise in the process of ecological compensation from a qualitative perspective. This paper tries to give a better explanation of the decision-making between government and enterprises from a quantitative perspective. The practical significance is that it can relieve the financial pressure of government and improve the efficiency of ecological compensation.

The implications of this study for managers are as follows: 1) under different ecological compensation coordination mechanisms, the government and enterprises can obtain different benefits, and only in the mode of cooperation between the government and enterprise can the two parties obtain the optimal benefits. 2) Government can establish an information sharing platform with enterprises to make a unified decision on the level of ecological compensation, and to improve the efficiency of government-enterprise cooperation. 3) At the same time increase the publicity of ecological compensation and increase the level of cost sharing for participating enterprise, so to encourage enterprise to actively carry out ecological compensation.

The main limitation of this method is that the determination of each parameter value is subjective to a certain extent, and the parameters will be determined through big data association analysis in subsequent studies. In addition, this paper only considers the positive impact of enterprise ecological compensation on the ecological environment, but does not consider the possible problems caused by the level of compensation of competing enterprise. The negative impact of enterprise dishonesty behaviors in the compensation process on its social reputation. Future research can be carried out from the above points.

## Author Contributions

**Conceptualization:** Hao Sun, Guangkuo Gao, Zonghuo Li.

**Data curation:** Hao Sun.

**Formal analysis:** Guangkuo Gao.

**Methodology:** Zonghuo Li.

**Writing – original draft:** Hao Sun.

**Writing – review & editing:** Hao Sun.

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
