## [Decision Letter · Decision Letter 0]

12 Apr 2021

PONE-D-21-07220

Research on the Cooperative Mechanism of Government and Enterprise for Basin Ecological Compensation Based on Differential Game

PLOS ONE

Dear Dr. Sun,

Thank you for submitting your manuscript to PLOS ONE. After careful consideration, we feel that it has merit but does not fully meet PLOS ONE’s publication criteria as it currently stands. Therefore, we invite you to submit a revised version of the manuscript that addresses the points raised during the review process.

We look forward to receiving your revised manuscript.

Kind regards,

Carlos Gracia-Lázaro

Academic Editor

PLOS ONE

Additional Editor Comments:

Please, pay attention to comments by both reviewers. In particular, Reviewer 1 exposes some very critical comments that deserve special attention as they affect the integrity of the research.

Journal Requirements:

3. Please amend the manuscript submission data (via Edit Submission) to include author Gao Guang-Kuo and Li Zong-Huo.

Reviewers' comments:

Reviewer's Responses to Questions

**Comments to the Author**

1. Is the manuscript technically sound, and do the data support the conclusions?

Reviewer #1: Partly

Reviewer #2: Partly

2. Has the statistical analysis been performed appropriately and rigorously? 

Reviewer #1: No

Reviewer #2: Yes

3. Have the authors made all data underlying the findings in their manuscript fully available?

Reviewer #1: Yes

Reviewer #2: Yes

4. Is the manuscript presented in an intelligible fashion and written in standard English?

Reviewer #1: Yes

Reviewer #2: Yes

5. Review Comments to the Author

Reviewer #1: In this work, the authors consider the role of enterprises into the ecological compensation process. They make use of the differential game theory to explore the impact of different compensation mechanisms on the revenue of government and enterprises. The paper may contain novel ideas and provide suggestive reference for the government decision-making.

1. However, I might be suspicious about the correctness of your results, since there are some obvious errors in the analysis solution of differential equations. For example, in Eq. (12) and (14), I do not agree with the results you got in Eq. (14) when substituting the expression V1and V2 into Eq. (12). The first term in Eq. (14) is lack of ‘R’ and the numerator in the last term should be lambda2 so that it could correspond to that in Eq. (12). Additionally, the solution b2 seems to be partially unfinished and I cannot obtain the equation (15) by substituting a1 and a2, where it seems a1 and b1 should not be the same. The quantities ‘Vb’ and ‘Vg’ should not appear in Eq. (22) and (24) and so on. To this end, I recommend authors should make a detailed check of the solutions for all the compensation mechanisms.

2. I also think the introduction part is rather lengthy and wordy about the advantages of involving the enterprises into the ecological compensation process. I would recommend that here authors should give some background knowledge of the model you used.

3. I suggest that the manuscript needs editing well since there are some grammatical errors, such as the disagreement of singular-plural.

Reviewer #2: The paper entitled “Research on the Cooperative Mechanism of Government and Enterprise for Basin Ecological Compensation Based on Differential Game” deals with actual and very interesting topic. It is quite interesting and informative to most readers of this field such as PhD students, and might have some interest by practitioners.

However, I have the following comments that hopefully help the authors improve their paper:

• The introduction section is mixed with a literature review. I suggest to the authors a section dedicated to literature review where should analyse the existing works in the way to show the gap in the literature compared to this work.

• The authors should convince the readers of this journal, that their contribution is so important.

• In relation to literature review, it would be better if authors can have a table comparing the closely related works and theories on various dimensions and clearly showing the contribution of the paper.

• I suggest that the authors add a research method diagram. This will provide a snapshot of the research steps followed and will help the reader in a clearer understanding of the paper.

• In chapter 5, dedicated to analysis of calculation examples, Is there any other source of information to support the parameters’ selection in the model?

• All the results provided in the paper should be compared with other approaches. In order to evaluate the robustness of the authors’ proposed.

• These issues deserve a deeper discussion. What are the main limitations of this approach? What are the implications for theory and practice? What are the managerial implications from this research? How decision or policy makers could benefit from this study.

• As usual a final thorough proof-reading is recommended.

I encourage the authors to think along those questions and to develop this work further along those lines.

6. PLOS authors have the option to publish the peer review history of their article (what does this mean?). If published, this will include your full peer review and any attached files.

Reviewer #1: No

Reviewer #2: No

---

## [Author Response · Author response to Decision Letter 0]

19 May 2021

Dear Editor and Reviewers:

On behalf of my co-authors, we thank you very much for giving us an opportunity to revise our manuscript and appreciate your positive and constructive comments very much. Those comments are all valuable and very helpful for revising and improving our paper, as well as the important guiding significance to our researches. We have studied comments carefully and made corrections which, we hope, have improved the quality of the paper to your satisfaction. 

The revisions have been highlighted across the text. Our responses to your valuable comments and the main corrections in the paper are as follows: 

Responds to Reviewer #1

Comment 1: I might be suspicious about the correctness of your results, since there are some obvious errors in the analysis solution of differential equations. For example, in Eq. (12) and (14), I do not agree with the results you got in Eq. (14) when substituting the expression V1and V2 into Eq. (12). The first term in Eq. (14) is lack of ‘R’ and the numerator in the last term should be lambda2 so that it could correspond to that in Eq. (12). Additionally, the solution b2 seems to be partially unfinished and I cannot obtain the equation (15) by substituting a1 and a2, where it seems a1 and b1 should not be the same. The quantities ‘Vb’ and ‘Vg’ should not appear in Eq. (22) and (24) and so on. To this end, I recommend authors should make a detailed check of the solutions for all the compensation mechanisms. We believe that these changes are in line with your requirements.

Response: Thank you for your comments. Following your comment, we are guilty of a basic calculation error. We have carefully examined all the compensation mechanism solutions in the paper and corrected the errors. In order to make the calculation process more clear, we refine the calculation of individual formulas in the paper, but it does not affect the conclusion in the paper. We have also replaced the confusing symbols in the article, for example, is replaced by, all the substitution symbols are marked.

Comment 2: I also think the introduction part is rather lengthy and wordy about the advantages of involving the enterprises into the ecological compensation process. I would recommend that here authors should give some background knowledge of the model you used.

Response: Thanks for your valuable comment. Following your comment, we have simplified the description of the advantages of enterprises participating in ecological compensation in the introduction, and added the background knowledge of differential game model in the introduction. 

The deletion of a lengthy part of the introduction as follows:

Ecological compensation system has been widely used in the world, its connotation includes two aspects: The first is to compensate for the ecological environment, mainly including the cost of water and soil conservation, greening and pollution control in the basin; The other is the compensation for participants, that is to the relevant subjects involved in ecological protection will be given certain incentives and preferences., such as land subsidies for residents who have returned farmland, tax incentives for green and energy-saving industries, etc.

The added background knowledge of the game model is as follows:

Differential game theory originated from the research on the pursuit of two parties in military confrontation carried out by the US Air Force in the 1950s. It is a combination of optimal control and game theory. It studies the continuous game of multiple players in a time-continuous system, in which the players try to optimize their independent goals, and eventually reach a Nash equilibrium over time. 

Comment 3: I suggest that the manuscript needs editing well since there are some grammatical errors, such as the disagreement of singular-plural.

Response: Thank you for your valuable suggestion. We have carefully checked the paper and corrected some grammatical errors in the paper.

Responds to Reviewer #2

Comment 1: The introduction section is mixed with a literature review. I suggest to the authors a section dedicated to literature review where should analyse the existing works in the way to show the gap in the literature compared to this work.

Response: Thanks for your good comment. Following your comment, we separately describe the introduction and literature review, and added some literature to enrich the literature review part of the paper.

Comment 2: The authors should convince the readers of this journal, that their contribution is so important.

Response: Thank you for your valuable suggestion. Following your comment, we have added the importance of the research. We believe that these changes are in line with your requirements. The corresponding content is copied as follows:

Watershed ecological compensation needs a large amount of funds. If only the payment is made by the government finance, it will cause great pressure on the government finance, and it is difficult to realize the sustainable compensation. Enterprise are the main cause of river basin pollution and the main beneficiary of river basin pollution. According to the principle that whoever damages shall restore and who benefits shall compensate, the enterprise shall bear the responsibility of compensation. Therefore, it is of great significance to bring polluting enterprises into the research framework of ecological compensation, and to discuss the decision-making behavior and influencing factors of government and enterprises in the process of ecological compensation.

Comment 3: In relation to literature review, it would be better if authors can have a table comparing the closely related works and theories on various dimensions and clearly showing the contribution of the paper.

Response: Thank you for your valuable comment. Following your comment, we have added a literature table to show the content and characteristics of the scholars' research, and also describe the differences between our research and the scholars' previous research. The corresponding content is shown in Table 1.

Comment 4: I suggest that the authors add a research method diagram. This will provide a snapshot of the research steps followed and will help the reader in a clearer understanding of the paper.

Response: Thanks for your valuable comment. Following your comment, We have added a research method diagram to describe the research process in detail, it can help the reader clearer understand the paper. The corresponding content is shown in Figure 2.

Comment 5: In chapter 5, dedicated to analysis of calculation examples, Is there any other source of information to support the parameters’ selection in the model?

Response: Thanks for your valuable comment. Following your comment, We explain

the sources of the parameters in the article. Since the ecological compensation mechanism proposed in this paper is a new operations management theory, direct real data cannot be obtained. Therefore, this paper combined with the relevant data in China Environmental Statistics Yearbook and Local statistical yearbook, make the parameter setting as practical as possible. The data in the China Environmental Statistics Yearbook and Local statistical yearbook are open access. Specific reference data are as follows:

China Environmental Statistics Yearbook (2020)

https://navi.cnki.net/KNavi/YearbookDetail?pcode=CYFD&pykm=YHJSD&bh=

Second, water environment; Tenth, environmental investment

China Environmental Statistics Yearbook (2019)

https://navi.cnki.net/KNavi/YearbookDetail?pcode=CYFD&pykm=YZGHW&bh=N2020070223

Environmental Planning Institute of the Ministry of Ecology and Environment: Ecological compensation and biodiversity conservation

Suzhou Yearbook (2020)

https://navi.cnki.net/KNavi/YearbookDetail?pcode=CYFD&pykm=YSZNJ&bh=

Selections of documents: The Office of the Suzhou Municipal Committee of the Communist Party of China (CPC) and the Office of the Suzhou Municipal People's Government issued the Notice on the Implementation of Opinions on Comprehensively Promoting and Implementing the Experience of the Pilot Ecological Compensation Mechanism in the Xin 'an River Basin

Finance Yearbook of Anhui Province (2020)

https://navi.cnki.net/KNavi/YearbookDetail?pcode=CYFD&pykm=YAHCZ&bh=

Support the battle against pollution: To support horizontal ecological compensation in the upper and lower reaches of the Xin 'an River Basin; Promote ecological compensation for water environment in Dabie Mountain Area; establish and improve mechanisms for compensating for ecological damage.

fiscal management: Anhui innovates to implement different types of ecological compensation mechanisms

Yearbook of Lu Quan (2020)

https://navi.cnki.net/KNavi/YearbookDetail?pcode=CYFD&pykm=YLUQU&bh=

Ecological poverty alleviation: ecological compensation to increase income

Financial Yearbook of Zhejiang Province (2019)

https://navi.cnki.net/KNavi/YearbookDetail?pcode=CYFD&pykm=YZJCZ&bh=N2020010090

Research Report Selection: Quzhou City's Practice and Reflection on Establishing the Whole Urban Upstream and Downstream Ecological Compensation Mechanism

Hefei Yearbook (2019)

https://navi.cnki.net/KNavi/YearbookDetail?pcode=CYFD&pykm=YHFNJ&bh=N2021040077

Ecological construction and environmental protection: Chaohu Lake management

Guiyang Yearbook (2019)

https://navi.cnki.net/KNavi/YearbookDetail?pcode=CYFD&pykm=YPAKF&bh=N2019120318

Water environmental management: ecological compensation for water pollution control

Xuancheng Yearbook (2019)

https://navi.cnki.net/KNavi/YearbookDetail?pcode=CYFD&pykm=YAHXC&bh=N2020030245

Pollution prevention and control: water environment ecological compensation

Comment 6: All the results provided in the paper should be compared with other approaches. In order to evaluate the robustness of the authors’ proposed.

Response: Thanks for your valuable comment. Following your comment, We compare the research method in this paper with other research methods and illustrate the advantages of this research method. We believe that these changes are in line with your requirements. The corresponding content is copied as follows:

In this paper, the differential game model is used to study the ecological compensation problem, which gets rid of the original research model from the qualitative perspective. Considering the influence of enterprise reputation on enterprise sales, and taking the enterprise sales as the decision variable, the dynamic process of decision-making between government and enterprise in the process of ecological compensation can be described more accurately.

Comment 7: These issues deserve a deeper discussion. What are the main limitations of this approach? What are the implications for theory and practice? What are the managerial implications from this research? How decision or policy makers could benefit from this study.

Response: Thanks for your valuable comment. Following your comment, We have supplemented the main limitations of this approach, the theoretical and practical implications of the study, the management implications of the study, and how decision makers might benefit from the study. We believe that these changes are in line with your requirements. The corresponding content is copied as follows:

The main limitation of this method is that the determination of each parameter value is subjective to a certain extent, and the parameters will be determined through big data association analysis in subsequent studies. 

The theoretical significance of this paper breaks away from the routine of previous scholars who studied the game between government and enterprises in the process of ecological compensation from a qualitative perspective, and tries to give a better explanation of the decision-making between government and enterprises from a quantitative perspective. The practical significance is that it can relieve the financial pressure of government departments and improve the efficiency of ecological compensation. 

The implications of this study for managers are as follows: under different ecological compensation coordination mechanisms, the government and enterprises can obtain different benefits, and only in the mode of cooperation between the government and enterprises can the two parties obtain the optimal benefits.

Government departments can establish an information sharing platform with enterprises to make a unified decision on the level of ecological compensation, and at the same time increase the publicity of ecological compensation and increase the level of cost sharing for participating enterprises, so as to encourage enterprises to actively carry out ecological compensation.

Comment 8: As usual a final thorough proof-reading is recommended.

Response: Thanks for your comment. Thorough proofreading has been conducted so that the grammatical, spelling, punctuation, and usage mistakes strewn across the manuscript have been rectified. 

Special thanks to you for your careful review and good comments, we to think along those questions and to develop this work further along those lines.

---

## [Decision Letter · Decision Letter 1]

8 Jun 2021

PONE-D-21-07220R1

Research on the Cooperative Mechanism of Government and Enterprise for Basin Ecological Compensation Based on Differential Game

PLOS ONE

Dear Dr. Sun,

Thank you for submitting your manuscript to PLOS ONE. After careful consideration, we feel that it has merit but does not fully meet PLOS ONE’s publication criteria as it currently stands. Therefore, we invite you to submit a revised version of the manuscript that addresses the points raised during the review process.

We look forward to receiving your revised manuscript.

Kind regards,

Carlos Gracia-Lázaro

Academic Editor

PLOS ONE

Journal Requirements:

Reviewers' comments:

Reviewer's Responses to Questions

**Comments to the Author**

1. If the authors have adequately addressed your comments raised in a previous round of review and you feel that this manuscript is now acceptable for publication, you may indicate that here to bypass the “Comments to the Author” section, enter your conflict of interest statement in the “Confidential to Editor” section, and submit your "Accept" recommendation.

Reviewer #1: All comments have been addressed

Reviewer #2: All comments have been addressed

2. Is the manuscript technically sound, and do the data support the conclusions?

Reviewer #1: Partly

Reviewer #2: Yes

3. Has the statistical analysis been performed appropriately and rigorously? 

Reviewer #1: Yes

Reviewer #2: N/A

4. Have the authors made all data underlying the findings in their manuscript fully available?

Reviewer #1: Yes

Reviewer #2: Yes

5. Is the manuscript presented in an intelligible fashion and written in standard English?

Reviewer #1: Yes

Reviewer #2: Yes

6. Review Comments to the Author

Reviewer #1: (No Response)

Reviewer #2: The manuscript has significantly improved as compared to the previous version. Indeed, the authors tried to improve it, and the main weaknesses are solved.

I am also satisfied with the responses and explanations given by the authors to my comments.

Thus, in my opinion, the manuscript is recommendable for publication.

7. PLOS authors have the option to publish the peer review history of their article (what does this mean?). If published, this will include your full peer review and any attached files.

Reviewer #1: No

Reviewer #2: No

---

## [Author Response · Author response to Decision Letter 1]

13 Jun 2021

Dear Editor and Reviewers:

We thank you very much for giving us an opportunity to revise our manuscript and appreciate your positive and constructive comments very much. We have studied comments carefully and made corrections which, we hope, have improved the quality of the paper to your satisfaction. The revisions have been highlighted across the text. Our responses to your valuable comments and the main corrections in the paper are as follows: 

Comment 1: Although authors have made most corrections for the compensation mechanism solutions, there are still some errors in the terms of writing. For example, in Eq. (2) and (24), the advertising effect coefficient is “α”, while in the other expressions, this parameter is replaced by “a”. I recommend authors should keep it consistent. In Eq. (22), I believe “2A-A” should be “2A-B”. Besides, I cannot get the results shown in Eqs. (29-30) when substituting the expressions (25), (27) and (28) into (24) and (26) except the numerator of the “L” should be “π1a” or “π1α” rather than “π1µ”. When substituting Eq. (42) into Eq. (41), I found it should be “+” instead of “-” in the first term of Eq. (44). In Eq. (47), the term “1/ρ” in Eq. (46) miss out. I cannot obtain the Eq. (48), since the factor α in front of π2 in Eq. (22) is lost. I highly recommend authors to double check all the expressions again.

Response: Thank you for your comments. Following your comment, we have carefully checked all the expressions in the paper and corrected the errors. The revisions have been highlighted across the text. 

Comment 2: I would suggest that the authors should address the captions of Figs. 4-6 in detail. The captions should include the interpretation of x-axis and y-axis, the use of parameters, and the implication of lines in each figure, etc. In addition, the text size of figures is a little bit small. Therefore, I recommend that each figure should be taken separately for clear visualization and analysis.

Response: Thanks for your valuable comment. Following your comment, we explained the meaning of the X-axis and Y-axis and the related parameters in the figure. In addition, we analyze each figure separately and explain the meaning of the lines in each figure. 

Comment 3: The authors have chosen only one set of parameters for the calculation analysis. I am curious to see how the results would change if other parameters 

were chosen. 

Response: Thank you for your valuable suggestion. Following your comment, we have added the sensitivity analysis table of related parameters, and the influence of the change of related parameters on the results is analyzed. We believe that these changes are in line with your requirements. 

Special thanks to you for your careful review and good comments, we hope that our modifications can meet your requirements.

---

## [Decision Letter · Decision Letter 2]

23 Jun 2021

PONE-D-21-07220R2

Research on the Cooperative Mechanism of Government and Enterprise for Basin Ecological Compensation Based on Differential Game

PLOS ONE

Dear Dr. Sun,

Thank you for submitting your manuscript to PLOS ONE. After careful consideration, we feel that it has merit but does not fully meet PLOS ONE’s publication criteria as it currently stands. Therefore, we invite you to submit a revised version of the manuscript that addresses the points raised during the review process.

Please, pay special attention to the issues raised by Referee 1, as this may be the last opportunity to correct them. Specify in your answer the specific changes you have made about the points specified by the reviewer.

We look forward to receiving your revised manuscript.

Kind regards,

Carlos Gracia-Lázaro

Academic Editor

PLOS ONE

Journal Requirements:

Additional Editor Comments (if provided):

Please, pay special attention to the issues raised by Referee 1, as this may be the last opportunity to correct them. Specify in your answer the specific changes you have made about the points specified by the reviewer.

Reviewers' comments:

Reviewer's Responses to Questions

**Comments to the Author**

1. If the authors have adequately addressed your comments raised in a previous round of review and you feel that this manuscript is now acceptable for publication, you may indicate that here to bypass the “Comments to the Author” section, enter your conflict of interest statement in the “Confidential to Editor” section, and submit your "Accept" recommendation.

Reviewer #1: All comments have been addressed

2. Is the manuscript technically sound, and do the data support the conclusions?

Reviewer #1: Yes

3. Has the statistical analysis been performed appropriately and rigorously? 

Reviewer #1: N/A

4. Have the authors made all data underlying the findings in their manuscript fully available?

Reviewer #1: Yes

5. Is the manuscript presented in an intelligible fashion and written in standard English?

Reviewer #1: Yes

6. Review Comments to the Author

Reviewer #1: (No Response)

7. PLOS authors have the option to publish the peer review history of their article (what does this mean?). If published, this will include your full peer review and any attached files.

Reviewer #1: No

---

## [Author Response · Author response to Decision Letter 2]

23 Jun 2021

Dear Editor and Reviewers:

 We thank you very much for your comments. We have studied comments carefully and made corrections which, we hope, have improved the quality of the paper to your satisfaction. The revisions have been highlighted across the text. Our responses to your valuable comments and the main corrections in the paper are as follows: 

 Thank you for your comments. Following your comment, we have carefully checked all the expressions in the paper and corrected the errors. The revisions have been highlighted across the text. 

 Special thanks to you for your careful review and good comments, we hope that our modifications can meet your requirements.

---

## [Editor Report · Decision Letter 3]

28 Jun 2021

Research on the Cooperative Mechanism of Government and Enterprise for Basin Ecological Compensation Based on Differential Game

PONE-D-21-07220R3

Dear Dr. Sun,

We’re pleased to inform you that your manuscript has been judged scientifically suitable for publication and will be formally accepted for publication once it meets all outstanding technical requirements.

Kind regards,

Carlos Gracia-Lázaro

Academic Editor

PLOS ONE
---

## [Editor Report · Acceptance letter]

14 Jul 2021

PONE-D-21-07220R3 

Research on the Cooperative Mechanism of Government and Enterprise for Basin Ecological Compensation Based on Differential Game 

Dear Dr. Sun:

I'm pleased to inform you that your manuscript has been deemed suitable for publication in PLOS ONE. Congratulations! Your manuscript is now with our production department. 

Kind regards, 

on behalf of

Dr. Carlos Gracia-Lázaro 

Academic Editor

PLOS ONE